# Long-Chain Polyunsaturated Fatty Acid Concentrations and Association with Weight Gain in Pregnancy

**DOI:** 10.3390/nu14010128

**Published:** 2021-12-28

**Authors:** Jerusa da Mota Santana, Marcos Pereira, Gisele Queiroz Carvalho, Djanilson Barbosa dos Santos, Ana Marlucia Oliveira

**Affiliations:** 1Center of Health Sciences, Universidade Federal do Recôncavo da Bahia, Avenida Carlos Amaral, R. do Cajueiro, 1015, Santo Antonio de Jesus 44574-490, Brazil; djanilsonb@gmail.com; 2Collective Health Institute, Universidade Federal da Bahia, Rua Basílio da Gama, s/n—Canela, Salvador 40110-040, Brazil; pereira.santosm@yahoo.com; 3Campus Avançado de Governador Valadares, Universidade Federal de Juiz de Fora, Governador Valadares 35010-17, Brazil; giseleqc@outlook.com

**Keywords:** pregnancy, omega-3 fatty acids, omega-6 fatty acids, pregnancy outcome, weight gain

## Abstract

Lower concentrations of omega-3 (ω-3) and higher concentrations of omega-6 (ω-6) have been associated with excess weight in adults; however, the information on this relationship in pregnancy remains in its infancy. This study aimed to investigate the association between plasma levels of ω-3 and ω-6 long-chain polyunsaturated fatty acids (PUFAs) and weight gain during the gestational period. This is a prospective cohort study involving 185 pregnant women registered with the prenatal services of a municipality in the northeast of Brazil. The dosage of the serum concentration of fatty acids and the anthropometric measurements were carried out at the baseline, and the women’s weight information in the first, second, and third trimesters was collected from their pregnancy cards. Serum fatty acids were determined with the help of gas chromatography. The response variable of this study is the latent variable weight gain in pregnancy, derived from three variables: gestational weight in the first, second, and third trimesters. The main exposure was the plasma concentrations of PUFAs. Structural equation modeling was used for the data analysis. The mean age of the pregnant women was 26.74 years old (SD: 5.96 years). Most of the women had not completed high school (84%) and had a low income (70.86%). It was observed that the ω-3 PUFAs, represented by ALA plasm (alpha-linolenic acid), DHA (docosahexaenoic acid), and the EPA/ALA ratio (eicosapentaenoic acid to alpha-linolenic acid ratio), were negatively associated with the weight gain during pregnancy construct (−0.20, −0.12, and −0.14, respectively). Meanwhile, the PUFAs represented by the ratio between the ω-6 category acids ARA and LA (arachidonic acid and linoleic acid) had a direct and positive association (0.22) with that construct. Excess maternal weight gain was associated with ω-3 and ω-6 plasma levels. The women with the greatest gestational weight gain were the ones that presented the highest ARA/LA ratio (ω-6) and the lowest plasma concentrations of ALA, DHA, and EPA/ALA ratio (ω-3).

## 1. Introduction

The current evidence indicates that the concentration of polyunsaturated fatty acids (PUFAs) has continuously decreased in Western diets due to the reduction in total fat and saturated fat in the total percentage of calories in the diet [1,2]. On the other hand, the intake of omega-6 (ω-6) fatty acids has increased and that of omega-3 (ω-3) fatty acids has decreased, resulting in a major increase in the proportion of ω-6 to ω-3 [3].

This change in the composition of fatty acids has been accompanied by a significant increase in the prevalence of excess weight and obesity. This may be explained by the excess ω-6 inhibiting the activation of the Δ-6-desaturase enzyme, which maintains a balance in the proportion of ω-6 to ω-3, and consequently the conversion of ω-3 into its active forms (EPA and DHA) [4]. Moreover, excess ω-6 (LA and ARA) has been associated with an increase in the inflammatory process and weight gain in individuals [1,3].

ω-3 (ALA, EPA, and DHA) has an anti-inflammatory effect, impeding the inflammation route [1], and it participates in the inhibitory process of dietary intake. It also controls the excessive storage of peripheral tissues such as adipose tissue [5] and improves the sensitivity and secretion of adipokines (adiponectin and leptin), hormones secreted by the adipose tissue and liver [6], thus contributing to a balanced body.

However, the results on the effects of ω-3 on adult body weight regulation remain inconsistent. Some studies record an association between consuming foods that are sources of ω-3 and reduced body weight [7,8,9], while others do not find any significant effect in any direction of association [10,11], independently of the individual’s gender, ethnicity, or age. Greater consumption of vegetable oils and margarines that are sources of LA, especially contained in processed and ultraprocessed foods, leads to a greater supply of these nutrients in the diet [2].

Investigations into the relationship between plasma levels of fatty acids and weight gain in pregnancy are scarce. Among the information available, it is possible to identify an association between excess weight in pregnancy and lower concentrations of ω-3 PUFA [12,13] and higher concentrations of ω-6 PUFA and a higher ω-6/ω-3 ratio [12]. Moreover, women who showed excessive gestational weight gain had higher concentrations of ω-6 PUFA (LA) and saturated fatty acids (myristic, palmitic, and stearic acid) [13] when compared with those of mothers how showed adequate weight gain.

Thus, it can be assumed that a high plasma concentration of ω-6 PUFA during pregnancy is a risk factor for excessive gestational weight gain, while a higher concentration of ω-3 PUFA protects against excess weight gain and may be promising for the prevention of excess weight and obesity in this and in subsequent life cycles. Therefore, this study aims to investigate the association between plasma levels of ω-3 and ω-6 acids and weight gain during the gestational period.

## 2. Materials and Methods

### 2.1. Study Design

This is a prospective and dynamic cohort study involving pregnant women registered with the prenatal services of 14 family healthcare units of the municipality of Santo Antônio de Jesus, in Bahia, Brazil. The study took place from August of 2013 to December of 2014. The follow-up lasted an average of nine months, including three three-month waves. Details of NISAMI studies can be found in previous studies [14,15].

### 2.2. Exclusion and Inclusion Criteria

The following were considered as eligible: clinically healthy pregnant women aged 18 or older, who were resident in the urban area of the municipality, with a gestational age of less than or equal to 34 weeks, confirmed by ultrasound, who received prenatal care at family healthcare units. The following women were excluded from the study: those with multiple pregnancies; those adhering to a vegan diet; those with renal, contagious, immunological, and metabolic diseases, as well as those with a history of HIV; those without ultrasound confirmation of the gestational age; and those who resided in the rural area of the municipality.

The women who fulfilled the eligibility criteria (*n* = 319) were invited to take part in the research and answered a closed questionnaire. Blood samples and anthropometric data were collected from 272 women. Thirteen were excluded for refusing to answer the food frequency questionnaire (FFQ), six were excluded for not answering the whole questionnaire, three were excluded for having a miscarriage, and 65 women were excluded for not having PUFA measurements. After these exclusions, 185 women were monitored during pregnancy, in the period from August of 2013 to December of 2014. The cohort flowchart is presented in Figure 1.

### 2.3. Assessment of the Consumption of Fatty Acids

At the baseline of the study, the blood collection was carried out for the dosage of the serum concentration of fatty acids, which was considered as the main exposure variable for weight gain.

On the day preceding the collection, the women were telephoned to reinforce the protocol guidelines for carrying out the exams, including fasting for 12 h and abstaining from engaging in physical activities in the 24 h and from consuming alcohol in the 72 h prior to the exam. In a clinical analysis laboratory in the city, 8 mL of venous blood were collected using a tube containing EDTA.

The plasma was separated from the other blood components by means of centrifugation at 2500 rpm for 15 min and frozen immediately afterward in liquid nitrogen until the analyses were carried out. To identify the plasma profile of PUFAs (ALA (linolenic acid), LA (linoleic acid), EPA (eicosapentaenoic acid), DHA (docosahexaenoic acid), and ARA (arachidonic acid)), first the lipids were extracted from 1000 μL of plasma using the Folch method (2:1 chloroform/methanol Folch solution) [16]. The derivatization stage was performed using the Hartman and Lago method [17].

The fatty acids were identified with the help of gas chromatography, using a Shimadzu^®^ gas chromatograph with a flame ionization detector. To record and analyze the chromatographs, the device was attached to a microcomputer, and the GC Solution program was used. The compounds were separated and identified in a Carbowax capillary column (30 m × 0.25 mm).

The fatty acids were identified by comparing the retention times of the esters from the samples with the FAME mix reference standard (Sigma-Aldrich^®^, Bellefonte, PA, USA).

### 2.4. Evaluation of Weight Gain in Pregnancy

The outcome of this study is the latent variable weight gain in pregnancy, which was constructed based on the measurements of gestational weight in the first, second, and third trimesters. The anthropometric measurements were carried out at the baseline by the project team and the other weight information was captured in the three trimesters. Pre-gestational weight was collected from the women’s pregnancy cards and their medical records. The measurements were carried out by technicians from the service who were duly trained by the project team.

Maternal weight was calculated using a model 31 Filizola^®^ mechanical balance with 150 kg capacity and 100 g sensitivity. Height was calculated using a Sanny^®^ stadiometer with 2000 cm capacity and 0.1 cm sensitivity.

The pre-gestational body mass index (BMI) was employed to classify anthropometric status [18]. BMI according to gestational age was used to evaluate the women’s anthropometric status throughout the follow-up period using the curve from Atalah et al. [19] as a reference. To evaluate weight gain in pregnancy, we considered the difference between the pre-gestational weight and the weights in each trimester of pregnancy. Total weight gain was classified according to the Institute of Medicine parameters [18].

Gestational age was calculated using the gestogram technique, which consists of marking on a calendar the first day and month of the last menstruation and the day and month of the current consultation and identifying the gestational week number indicated on the gestogram. This was subsequently confirmed by the ultrasound exam and noted on the pregnant woman’s card.

### 2.5. Assessment of Other Variables

The following socioeconomic, demographic, reproductive, and lifestyle variables were obtained using a closed questionnaire at the baseline of the study: age (years), gestational week, pre-gestational BMI (in kg/m^2^), parity (number of children), education (years of study), family income (total income), number of residents (total residents in the household), smoking (yes/no), and alcohol consumption (yes/no).

### 2.6. Data Management and Statistical Analyses

The sample was calculated based on a 48.1% prevalence of excess weight among the pregnant women in the municipality where the study was conducted [20], a 3% sampling error, and 87% power.

First the descriptive analysis was carried out, adopting the mean and standard deviation for the quantitative variables and proportion for the qualitative ones. We carried out the bivariate analysis and Student’s *t*-test for independent samples with the aim of evaluating the difference of means of the sociodemographic, demographic, obstetric, and nutritional variables according to weight gain in pregnancy.

To evaluate the association between plasma concentrations of ω-3 and ω-6 and weight gain in pregnancy and other socioeconomic, demographic, and obstetric variables, we employed SEM [21]. In this model, the latent variable is represented by a circle and the observable ones are represented by rectangles (Figure 2). We evaluated the direct, indirect, and total associations of the relationships studied using the standardized coefficients (SCs) and these were interpreted according to Kline [22]: small effect (SC values close to 0.10 and −0.10), medium effect (SC values of 0.30 and −0.30), and strong effect (SC values > 0.50 and >−0.50).

To evaluate the goodness of fit of the model, we employed Bentler’s comparative fit index (CFI) and the root mean square error of approximation (RMSEA); the values of these indicators were 0.76 and 0.0001, respectively, indicating model adequacy. We employed the Stata software, version 12, for the statistical analyses.

## 3. Results

### 3.1. Cohort Characteristics

The sample was composed of 250 women (corresponding to 750 observations) with a mean age of 26.74 years old (SD: 5.96 years), 84% of which had not completed high school and 70.80% of whose income was less than or equal to two minimum wages. The mean age (*p* = 0.81) and consumption of monounsaturated fatty acids (*p* = 0.80) and polyunsaturated fatty acids (0.50) did not differ between the segment and losses.

At the start of pregnancy, 40% of the women were overweight or obese, and at the end of this process, 35.20% of them were overweight or obese.

Table 1 presents the sociodemographic and nutritional characteristics according to weight gain throughout pregnancy. It was observed that the women with excessive weight gain were older and had a higher pre-gestational BMI (PGBMI), already being classified as overweight, and they were also the ones with the highest number of people residing in their homes.

### 3.2. Association between Dietary Intake and Weight Gain

The results of the SEM between the plasma levels of PUFAs and maternal weight gain throughout pregnancy are presented in Table 2 and Figure 2. All the indicator variables employed to form the latent construct presented high factor loadings (>0.90), with the “weight_t2” variable (pregnant woman’s weight in the second trimester) being the one that contributed most with a factor loading of 0.98 (Table 2; Figure 2).

The direct association between plasma levels of PUFAs and the “gest_weight” construct, adjusted by the pre-gestational BMI, maternal age, interpartal interval, years of schooling, maternal alcohol consumption, and gestational week, varied according to the class of PUFA (ω-3 and ω-6). It was observed that the ω-3 represented by ALAplasm, DHA, and the EPA/ALA ratio were directly and negatively associated with weight gain during pregnancy (−0.20, −0.12, and −0.14, respectively), with it being indicated that the women with a higher rate of gestational weight gain had lower plasma concentrations of ALA and DHA.

Meanwhile, the PUFAs represented by the ratio between the ω-6 category acids (ARA/LA) had a direct and positive association (0.22) (Figure 2) with weight gain throughout the follow-up, with it being indicated that a higher plasma ARA/LA ratio was associated with greater weight gain during the gestational cycle. All the effects were statistically significant (Table 2). The plasma levels of LA and ARA (ω-6) and EPA (ω-3) in isolation did not have a statistically significant association with gestational weight gain.

The indirect association of the PUFAs in the gestational weight gain construct (“gest_weight”) was estimated by multiplying the standardized coefficient of PGBMI by the plasma levels of PUFAs and by the coefficients of effect of the plasma levels of PUFAs in the construct (Table 2). It was observed that despite the indirect associations between the PUFAs and gestational weight gain being small, they were statistically significant when intermediated by the PGBMI (ALA: −0.026; ARA/LA: 0.052; EPA/ALA: −0.025), indicating that a higher BMI before pregnancy can raise the plasma concentrations of ω-6 sub-products and these have an influence on increasing gestational weight (Table 3).

The evaluation of the direct effect of PGBMI on the plasma levels of PUFA indicated that PGBMI had a direct negative effect on the EPA/ALA ratio during pregnancy and a direct positive effect on the ARA/LA ratio; that is, the higher the PGBMI, the higher the serum concentrations of ARA and LA and the lower the concentrations of EPA and ALA (Table 2).

The total effect (sum of the direct and indirect effects) of plasma PUFAs on gestational weight gain presented a similar pattern to the one identified for the direct effect. Thus, it was negative for ALA (−0.2026), DHA (−0.1236), and EPA/ALA (−0.12564), which are considered to be ω-3 category fatty acids, and positive for ARA/LA (0.272), from the ω-6 category. A more expressive direct association was noted between the PUFAs and the gestational weight construct, with a small indirect effect (Table 3).

In the interpartal period, age, alcohol consumption, and gestational age were variables that had a direct positive association with the gestational weight gain construct, and schooling presented a direct negative association. Thus, the women who saw a high increase in gestational weight were the oldest ones, those who consumed the most alcohol, those who had a longer interval between births, and those with a lower educational level.

Therefore, it is observed that the women with the greatest gestational weight gain were the ones who presented the highest ARA/LA ratio (ω-6 products) and the lowest plasma concentrations of ω-3 products (ALA, DHA, and EPA/ALA ratio).

## 4. Discussion

In this investigation, we observed high excessive weight gain in pregnancy (35.2%). The excessive weight gain was associated with ω-3 and ω-6 series PUFA plasma levels. The women with the greatest weight gain during pregnancy were the ones who presented the highest ARA/LA ratio (ω-6) and the lowest plasma concentrations of ALA, DHA and EPA/ALA ratio (ω-3). It was also recorded that the women who began their pregnancy with a higher BMI were the ones who presented the lowest EPA/ALA ratio and the highest ARA/LA ratio during pregnancy.

Thus, our results reveal an association between ω-3 and ω-6 and weight gain in the women throughout pregnancy. It was noted that the fractions of ω-3 fatty acids (ALA, DHA, and EPA/ALA ratio) regulated weight gain throughout pregnancy, promoting weight increases within the recommended limits, while the derivatives of the ω-6 class (ARA/LA ratio) acted as risk factors, raising weight throughout pregnancy.

The results of this study can be added to those recorded by other investigations of pregnant Brazilian women [12] and those from other countries [13,23]. In these studies, women with excess gestational weight had lower plasma concentrations of ω-3 [12,13], lower EPA and DHA percentages [12], and higher ω-6 concentrations and a higher ω-6/ω-3 ratio [12]. Moreover, greater gestational weight gain was associated with higher concentrations of ω-6 PUFA (LA) and saturated fatty acids (myristic, palmitic, and stearic acid) [13] when compared with the parameters of mothers who had adequate weight gain during pregnancy.

Some physiological mechanisms are proposed to explain the relationship between plasma concentrations of PUFAs and the influence on body weight; however, it should be stressed that these theoretical pathways are not totally elucidated and still require methodologically robust investigations that can trace a consistent cause and effect relationship.

The available evidence derived from animal and human studies indicates that the main endogenous agonists of the endocannabinoid system are PUFA derivatives, especially arachidonic acid (ω-6). This acid activates cannabinoid receptors (CB1), which via the hypothalamic pathway activate orexigenic signals and promote an increase in appetite, consequently stimulating greater food consumption [24,25] and increasing the individual’s weight, as well as acting as a pro-inflammatory stimulator by activating the production of substances such as 2-series prostaglandins (PGE2) and 4-series leukotrienes (LTB4) [26,27]

The endocannabinoid system controls and regulates dietary behavior at both a central level (acting over the hypothalamus and nucleus accumbens) and peripheral level (adipose tissue, liver, muscle, and gastrointestinal tract). At the central level via the mesolimbic system, it connects the ventral tegmental area (VTA) to the pre-frontal cortex and to the limbic system through the amygdalas, the hippocampus, and the nucleus accumbens and they activate reward circuits, stimulating the consumption of foods that favor a feeling of pleasure and appetite induction [24]. At a peripheral level, it promotes increased fat accumulation and adipogenesis [24,28]. Furthermore, nutritional and metabolic imbalances in pregnancy increase the supply of nutrients to the feto-placental unit and lead to excessive fetal growth and fat mass deposition [29].

Meanwhile, ω-3 (ALA, EPA, and DHA) inhibits the inflammation route (Patterson et al., 2012; Wall et al., 2010), acting as an antagonist of the endocannabinoid system, blocking the cannabinoid receptors (CB1) and inhibiting orexigenic signals, which interferes in the reduction in dietary intake. In addition to controlling the quantity of peripheral tissue stored, such as adipose tissue [5,24], these acids improve the sensitivity and secretion of adipokines (adiponectin and leptin) by the adipose tissue and liver [6,30], acting in the modulation of lipids, and in the central nervous system (CNS), reducing lipogenesis [5,26].

The association between ω-3 (EPA and DHA) and body weight control observed in some studies has stimulated the development of more controlled research with robust methodological designs to investigate the contribution of ω-3 to reducing body weight [31]; however, the results of the studies remain controversial.

Given the protective effects of ω-3 in controlling weight and reducing the risk of developing diseases, this is revealed to be promising in the prevention and treatment of excess weight and obesity in populations [26,27].

In this study, a higher PGBMI was associated with a lower plasma concentration of ω-3, represented by the EPA/ALA ratio, and a higher concentration of ω-6, represented by the ARA/LA ratio, which may indicate that the nutritional conditions before pregnancy can influence the nutritional conditions in the gestational cycle.

A similar result to this study was observed in 129 pregnant women in the United States [23]. The women who were obese before pregnancy had lower concentrations of DHA, a 3-series PUFA, when compared to women with adequate weight.

### Strengths and Limitations

The results of this study contribute to understanding the indiscriminate increase in the occurrence of excess weight and obesity in pregnancy, explaining the worrying epidemiological situation, as the available knowledge indicates that an inadequate maternal nutritional status can influence the genome and promote greater chances of neonatal complications and non-transmissible chronic diseases in subsequent life stages of the child [32]. Gestational obesity is also associated with a higher risk of gestational diabetes, pregnancy-specific hypertensive syndromes [33], a low birth weight, and fetal macrosomia [34].

It should be highlighted that this study presents some limitations. The first concerns the inclusion of pregnant women residing in the urban area of the municipality due to the difficulty of accompanying these women in the extensive rural area. Second, the maternal PUFA data were unfortunately only measured at the baseline, due to difficulties in collecting blood from this group, and we performed the measurement of PUFAs in the serum, which is sensitive to maternal nutrition before the measurement [35]. However, guidelines on fatty acid consumption were provided at the time of the participant’s inclusion in the cohort. In addition, we investigated the consumption and supplementation of fatty acids at the pregnant women’s bedtime using a standardized instrument, which may reduce bias related to PUFA intake.

Moreover, some weight gain data were collected from the women’s pregnancy cards or their prenatal patient records at the healthcare units. However, collection teams and professionals were trained for the anthropometric evaluation and these teams received calibrated scales. The use of SEM is another strong point, as it was possible to include associated factors that mediated the relationship investigated.

## 5. Conclusions

Our results reveal that women with excess pre-gestational weight and those who gained more weight in pregnancy presented higher plasma fractions of ω-6 and lower fractions of ω-3, and so it can be assumed that the deregulation of weight in women in this period of high physiological vulnerability may also be associated with an imbalance in the ω-3: ω-6 ratio. Thus, this cohort aggregates new information about the implications of weight gain during pregnancy and its relationship with polyunsaturated fatty acids.

Despite the relevance of the results of this study, the replication of these findings in other populations of pregnant women is necessary to better understand the phenomenon and produce scientific evidence to support strategies for guiding adequate ω-3 consumption in pregnant women.

## Figures and Tables

**Figure 1 nutrients-14-00128-f001:**
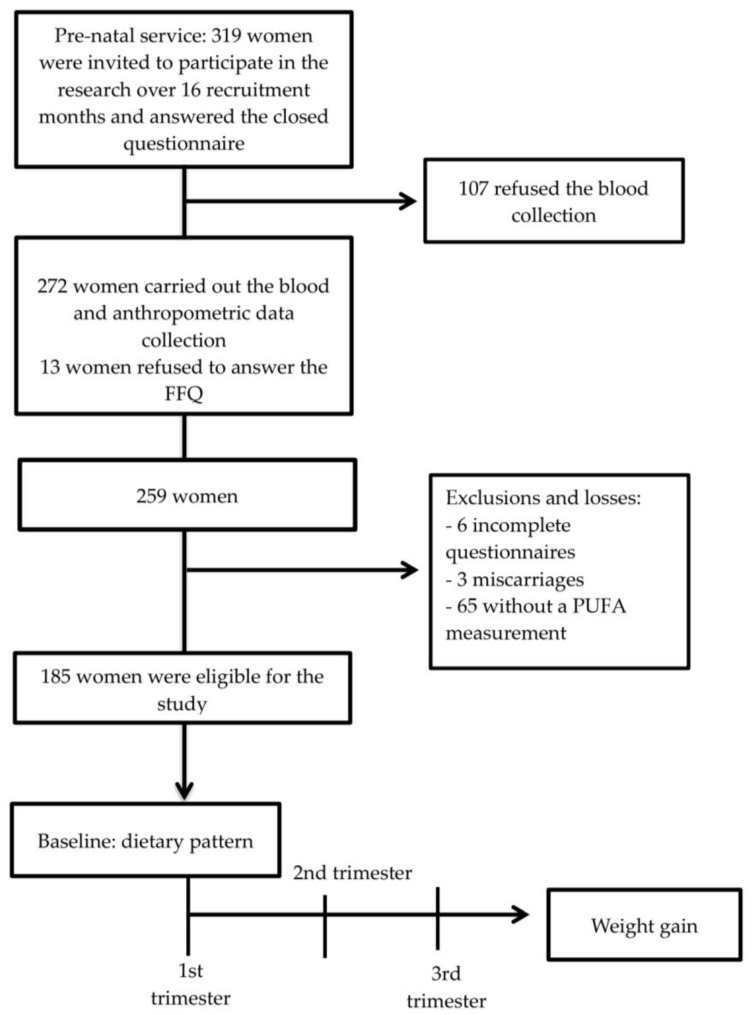
Flowchart of cohort of 185 pregnant women treated in the prenatal service at primary healthcare units in Santo Antônio de Jesus, Bahia, Brazil, 2017.

**Figure 2 nutrients-14-00128-f002:**
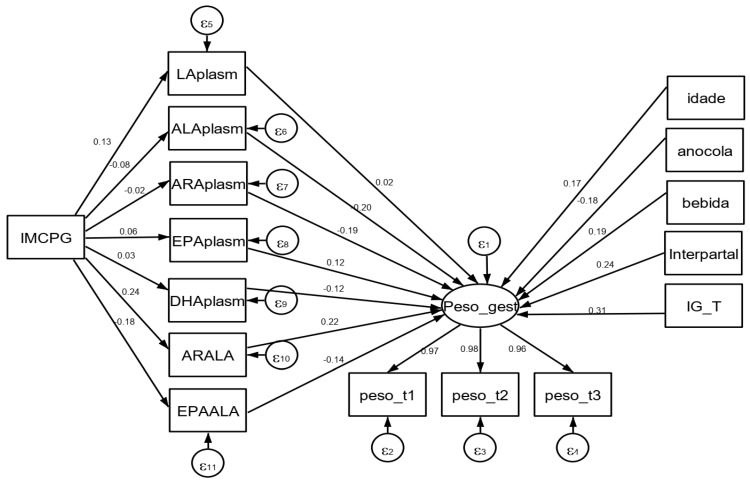
Structural equation modeling with the gestational weight construct and observable variables. ALA: linolenic acid; LA: linoleic acid; EPA: eicosapentaenoic acid; DHA: docosahexaenoic acid; ARA: arachidonic acid; EPA/ALA: eicosapentaenoic acid to linolenic acid ratio; ARA/LA: arachidonic acid to linoleic acid ratio; IMCPG: Pre-gestational body mass index. The construct that represents the weight gain throughout pregnancy (“gest_weight”) was built based on the indicator variables weight_t1 (weight in the first trimester), weight_t2 (weight in the second trimester), and weight_t3 (weight in the third trimester).

**Table 1 nutrients-14-00128-t001:** Sociodemographic and nutritional characteristics according to weight gain throughout pregnancy. Santo Antônio Jesus, Bahia, 2013–2014.

Characteristics	Expected Gestational Weight Gain		Excess Gestational Weight Gain		*p* Value ^1^
	Mean	SD	Mean	SD	
Maternal age	26.32	5.99	27.53	5.82	0.0038
Years of schooling	10.54	3.04	10.71	2.69	0.2116
Family income	1498.30	200	1472.95	200	0.1028
Number of inhabitants	3.000	1.51	4.000	1.55	0.0084
Gestational week	16.58	7.13	16.69	6.64	0.0018
PG BMI	23.84	4.66	25.33	4.74	0.0000
Plasma ALA	1.83	1.74	1.56	1.06	0.9851
Plasma LA	13.59	5.77	12.78	4.71	0.9614
Plasma EPA	2.68	2.12	2.47	2.14	0.8888
Plasma DHA	2.65	2.14	2.52	2.07	0.7009
Plasma ARA	1.70	1.29	1.65	1.29	0.0966
EPA/ALA	1.85	1.50	2.05	2.59	0.9838
ARA/LA	0.14	0.10	0.12	0.09	0.8420

ALA: linolenic acid; LA: linoleic acid; EPA: eicosapentaenoic acid; DHA: docosahexaenoic acid; ARA: arachidonic acid; EPA/ALA: eicosapentaenoic acid to linolenic acid ratio; ARA/LA: arachidonic acid to linoleic acid ratio. ^1^ Student’s *t*-test for independent samples. *n* = 185: 555 observations.

**Table 2 nutrients-14-00128-t002:** Structural equation modeling for the association between plasma levels of polyunsaturated fatty acids and maternal weight throughout pregnancy. Santo Antonio de Jesus, Bahia, 2013–2014.

Effects	Standardized Coeff.	*p* Value	IC 95%
Gestational weight ← Weight_T1	0.973	0.000	0.964–0.982
Gestational weight ← Weight_T2	0.983	0.000	0.976–0.991
Gestational weight ← Weight_T3	0.961	0.000	0.950–0.972
Gestational weight ← ALAplasm	−0.200	0.005	−0.340–−0.106
Gestational weight ← DHAplasm	−0.120	0.004	−0.236–−0.037
Gestational weight ← EPA/ALA	−0.140	0.050	−0.275–−0.080
Gestational weight ← ARA/LA	0.220	0.011	0.105–0.3948
Gestational weight ← Age	0.169	0.002	0.042–0.297
Gestational weight ← Interpartal	0.248	0.001	0.126–0.3680
Gestational weight ← Years of schooling	−0.180	0.002	−0.298–−0.062
Gestational weight ← GW	−0.313	0.003	0.200–0.426
Gestational weight ← Alcohol	0.197	0.001	0.085–0.3099
PGBMI ← EPA/ALA	−0.185	0.017	−0.337–−0.034
PGBMI ← ARA/LA	0.237	0.014	0.08–0.425
Goodness of fit indicators: RMSEA: 0.0001/CFI: 0.76			

ALA: linolenic acid; LA: linoleic acid; EPA: eicosapentaenoic acid; DHA: docosahexaenoic acid; ARA: arachidonic acid; RMSEA: root mean square error of approximation; CFI: comparative fit index. *n* = 185: 555 observations

**Table 3 nutrients-14-00128-t003:** Direct, indirect, and total effect of the PUFA concentrations on the weight gain construct intermediated by PGBMI. Santo Antônio de Jesus, Bahia, 2013–2014.

	Direct Effect	Indirect Effect	Total Effect
Gestational weight ← ALAplasm ← PGBMI	−0.200	−0.016	−0.216
Gestational weight l ← LAplasm ← PGBMI	0.020	0.0026	0.0226
Gestational weight ← DHAplasm ← PGBMI	−0.120	−0.0036	−0.1236
Gestational weight ← EPA/ALA ← PGBMI	−0.140	−0.0252	−0.1652
Gestational weight ← ARA/LA ← PGBMI	0.220	0.0528	0.2728
Gestational weight l ← ARAplasm ← PGBMI	−0.190	0.0038	−0.1938
Gestational weight ← EPAplasm ← PGBMI	0.120	0.0072	0.1272

ALA: linolenic acid; LA: linoleic acid; EPA: eicosapentaenoic acid; DHA: docosahexaenoic acid; ARA: arachidonic acid; EPA/ALA: eicosapentaenoic acid to linolenic acid ratio; ARA/LA: arachidonic acid to linoleic acid ratio. *n* = 185: 555 observations.

## Data Availability

The study did not report any data.

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
