# Peer review of "Long-Chain Polyunsaturated Fatty Acid Concentrations and Association with Weight Gain in Pregnancy"

_nutrients, 2021, doi:10.3390/nu14010128_

Round 1

Reviewer 1 Report

The manuscript submitted for publication by Da Mota Santana eat al., is investigating the effect of dietary omega-3/omega-6 ratio on weight status over the pregnancy period in pregnant women. The manuscript is well written and the presentation is really sound and well crafted. The study, materials and methods are described very well. The manuscript is structured logically and it flows well with an easy read for the reader. Some improvement in English (style, grammar and syntax) is recommended, possibly by an English native speaker for optimization of expression schemes. 

The reviewer would like to offer a few further points for the improvement of the manuscript. 

  1. What are the confounding factors potentially associated with the population? Could the authors comment on that?
  2. Did the authors consider supplements in their analyses. Often times women during pregnancy may take supplements including fish oil supplements for omega-3/6 fatty acid supplementation.
  3. Consider providing some more information as per the site of the study (characteristics of the location from which participants originate).
  4. Consider including a separate section on strengths and limitations of the study. 

Overall nice work!

Author Response

The manuscript submitted for publication by Da Mota Santana et al. investigates the effect of the dietary omega-3/omega-6 ratio on weight status over the pregnancy period in pregnant women. The manuscript is well written and the presentation is really sound and well crafted. The study, materials and methods are described very well. The manuscript is structured logically and it flows well, providing an easy read for the reader. Some improvement in English (style, grammar and syntax) is recommended, possibly by a native English speaker for optimization of expressions. 

The reviewer would like to offer a few further points for the improvement of the manuscript. 

  1. What are the confounding factors potentially associated with the population? Could the authors comment on these?
  2. Did the authors consider supplements in their analyses? During pregnancy women may often take supplements, including fish oil supplements, for omega-3/6 fatty acid supplementation.
  3. Consider providing some more information regarding the location of the study (characteristics of the location from which the participants originate).
  4. Consider including a separate section on strengths and limitations of the study. 

Overall, nice work!

Authors: We adjusted the article, including limitations and reviewing the entire text.

The statistical model is complex and performs several simultaneous adjustments.

Reviewer 2 Report

Jerusa da Mota Santana and colleagues in “Long-chain polyunsaturated fatty acids concentrations and association with weight gain in pregnancy” [Nutrients-1493719] reported nutritional data from the Brazilian study cohort NISAMI (Pereira-Santos et al., 2019; Santana et al., 2020).

The authors reported here maternal serum PFA profiles and anthropometric measures at baseline and Gestational Weight Gain (GWG) at I-II-III trimester of pregnancy. They hypothesized a link between these data, which was investigated with a Structural Equation Modeling, a relatively new and quite sophisticated statistical analysis.

The focus of this study is interesting and appealing.

The shift to Western diets is currently a matter of public concern. A typical consequence of Western nutritional patterns is represented by a major intake of omega-6 (ω-6) fatty acids opposite to a minor intake of omega-3 (ω-3) fatty acids, thus resulting in an increased ω-6 to ω-3 proportion. It is known that ω-3 has an anti-inflammatory effect, but its mechanism of action on body weight regulation remains still unclear.

There are some unclear points and limits about this study.

Although all the anthropometric measures, including GWG, were recorded at I-II-III trimester of pregnancy (indicated as time points T in the text), the maternal PUFAs data were unfortunately measured only at baseline.

According to the rationale clearly expressed in the paper, the aim was to investigate the relationships between GWG and PUFA (ω-3 and ω-6 acids) ‘during the gestational period’. It should have been appropriate to dose maternal serum fatty acids at the end of pregnancy (III Trimester or close to delivery) or even better at each time point of the study, in order to truly understand the link between maternal diet, nutritional and lifestyle habits, and pregnancy through the evaluation of some anthropometric measures (GWG and BMI).

Moreover, the gestational timing assigned to the baseline must be clarified. In the method section it is reported that ‘healthy pregnant women with a gestational age of less than or equal to 34 weeks’ were enrolled in the study, while Table 1 reports a gestational age between the I° and the II° trimester.

The analysis of long-chain polyunsaturated fatty acids in serum usually provides indications of the diet in the last few days, being sensitive to maternal nutrition up to 5-10 days before the dosage. A fatty acids analysis in red blood cells would have provided a more lasting dosage, close to red blood cells’ half-life (180 days). The Authors should comment on this choice and disclose it in the limits of the study.

Concerning the results, it would be important to know the caloric intake of the pregnant women at the time of the FFQ. Obviously gestational weight gain is strongly influenced by total caloric intake, therefore the question is: how was the caloric intake compared to the intake of the omega-6/omega-3? which is the variable influencing GWG? a multivariate analysis would be needed.

The Discussion is not always focused as it spans from adult studies to the pregnant population with some confusion. The former part could be reduced while more attention shouls be given to the pregnant population. Moreover, the Authors should better explain what this study adds to the already a vailable literature on the subject.

The paper is generally well written concerning the English language. The narrative language is clear.

MINOR COMMENTS:

  • Lines 138-140: please insert the brand of the cited instruments;
  • Page 5: please fix the 3.2 Paragraph’s title. I will suggest to move it to page 6;
  • Line 208: please use a capital letter for “weight-T2” as it appears in all the paper text.
  • Discussion, page 9, first paragraph: it would interesting to quote a recent paper by Ruotolo et al (European Gynecology and Obstetrics. 2021; 3(3):125-128) on the role of maternal flavor for pregnancy outcomes

Author Response

In “Long-chain polyunsaturated fatty acid concentrations and association with weight gain in pregnancy” [Nutrients-1493719], Jerusa da Mota Santana and colleagues reported nutritional data from the Brazilian NISAMI cohort study (Pereira-Santos et al., 2019; Santana et al., 2020).

The authors reported maternal serum PUFA profiles and anthropometric measures at the baseline and Gestational Weight Gain (GWG) at the I-II-III trimesters of pregnancy. They hypothesized a link between these data, which was investigated with Structural Equation Modeling, a relatively new and quite sophisticated statistical analysis technique.

The focus of this study is interesting and appealing.

The shift to Western diets is currently a matter of public concern. A typical consequence of Western nutritional patterns is a higher intake of omega-6 (ω-6) fatty acids and a lower intake of omega-3 (ω-3) fatty acids, thus resulting in an increased ω-6 to ω-3 ratio. It is known that ω-3 has an anti-inflammatory effect, but its mechanism of action on body weight regulation still remains unclear.

There are some unclear points and limitations regarding this study.Although all the anthropometric measures, including GWG, were recorded at the I-II-III trimesters of pregnancy (indicated as time points T in the text), the maternal PUFA data were unfortunately only measured at the baseline. According to the rationale clearly expressed in the paper, the aim was to investigate the relationships between GWG and PUFA (ω-3 and ω-6 acids) ‘during the gestational period’. It would have been appropriate to dose maternal serum fatty acids at the end of pregnancy (III trimester or close to delivery) or even better at each time point of the study, in order to truly understand the link between maternal diet, nutritional and lifestyle habits, and pregnancy through the evaluation of some anthropometric measures (GWG and BMI).

Authors:  It was not possible to dose serum fatty acids in each gestionational trimester due to the financial limitation of the cohort, however we should highlight that there is not much difference in the serum fatty acid concentrations during the first, second, and third trimesters of pregnancy. So, for that reason, a gestional age < 34 weeks cut-off point was adopted. Moreover, for most of the pregnant women in this cohort (59.1%), the serum fatty acid measurements were carried at the end of the second trimester of pregancy.

Moreover, the gestational timing assigned to the baseline must be clarified. In the method section it is reported that ‘healthy pregnant women with a gestational age of less than or equal to 34 weeks’ were enrolled in the study, while Table 1 reports a gestational age between the I and the II trimesters.

Authors:   The cut-off point adopted for entry into the study was a gestational age <= 34 weeks. Thus, 59.1% of the women had their serum fatty acid evaluation in their second trimester, 32.7% had it in their first trimester, and 8.2% had it in their third trimester.

The analysis of long-chain polyunsaturated fatty acids in serum usually provides indications of the diet in the last few days, being sensitive to maternal nutrition up to 5-10 days before the dosage. A fatty acids analysis in red blood cells would have provided a more lasting dosage, close to red blood cells’ half-life (180 days). The Authors should comment on this choice and disclose it in the limitations of the study.

Concerning the results, it would be important to know the caloric intake of the pregnant women at the time of the FFQ. Obviously gestational weight gain is strongly influenced by total caloric intake, therefore the question is: how did the caloric intake compare to the intake of omega-6/omega-3? What is the variable influencing GWG? A multivariate analysis would be needed.

Authors:   Calorie intake did not enter into the purpose of the study. The FFQ employed was quali-quantitative, but the focus of the study was on evaluating the consumption frequency of the dietary groups. Moreover, in the dietary intake analysis adopted, the factor analysis is robust and manages to capture the complexity of the diet beyond the nutrient.

The Discussion is not always focused as it spans from adult studies to the pregnant population with some confusion. The former part could be reduced while more attention should be given to the pregnant population. Moreover, the Authors should better explain what this study adds to the literature already available on the subject.

Authors:   We exclude some paragraphs about W-3 supplementation in adults.

In addition, the findings have been improved.
